# Ion-Pair Compounds of Strychnine for Enhancing Skin Permeability: Influencing the Transdermal Processes In Vitro Based on Molecular Simulation

**DOI:** 10.3390/ph15010034

**Published:** 2021-12-27

**Authors:** Lili He, Di Xiong, Lan Ma, Yan Liang, Teng Zhang, Zhiming Wu, Huaibo Tang, Xuewen Wu

**Affiliations:** 1Department of Pharmacy, School of Chemistry, Xiangtan University, Xiangtan 411105, China; helily1205@163.com (L.H.); Malan1031@126.com (L.M.); liangyan126wy@126.com (Y.L.); zhangteng765721@163.com (T.Z.); 2Department of Pharmaceutical Engineering, School of Chemical Engineering, Xiangtan University, Xiangtan 411105, China; xiongdi@xtu.edu.cn (D.X.); xdwuzm@xtu.edu.cn (Z.W.)

**Keywords:** strychnine, ion-pair, transdermal, molecular simulation

## Abstract

This research aimed to explore how Strychnine (Str) ion-pair compounds affect the in vitro transdermal process. In order to prevent the influence of different functional groups on skin permeation, seven homologous fatty acids were selected to form ion-pair compounds with Str. The in vitro permeation fluxes of the Str ion-pair compounds were 2.2 to 8.4 times that of Str, and Str-C_10_ had the highest permeation fluxes of 42.79 ± 19.86 µg/cm^2^/h. The hydrogen bond of the Str ion-pair compounds was also confirmed by Fourier Transform Infrared (FTIR) Spectroscopy, Nuclear Magnetic Resonance (NMR) Spectroscopy and molecular simulation. In the process of molecular simulation, the intercellular lipid and the viable skin were represented by ceramide, cholesterol and free fatty acid of equal molar ratios and water, respectively. It was found by the binding energy curve that the Str ion-pair compounds had better compatibility with the intercellular lipid and water than Str, which indicated that the affinity of Str ion-pair compounds and skin was better than that of Str and skin. Therefore, it was concluded that Str ion-pair compounds can be distributed from the vehicle to the intercellular lipid and viable skin more easily than Str. These findings broadened our knowledge about how Str ion-pair compounds affect the transdermal process.

## 1. Introduction

Strychnine (Str), as shown in Figure 1, is a traditional medicine comprised of spinal cord stimulant that is usually employed to treat hemiplegia and amblyopia by subcutaneous injection [1,2]. However, Str can excite the spinal cord reflex, cause tonic spasms and even lead to death caused by paralysis of the respiratory muscles [2,3]. Since Str has a very narrow therapeutic index, it is also known as a dangerous poison in spite of its therapeutic values. Some patients are prone to present poisoning symptoms during subcutaneous injection of a clinical dose because the peak blood concentration of Str subcutaneous injection can be reached in fifteen minutes [2,4].

The zero-order absorption can be easily achieved by a transdermal drug delivery system (TDDS), which can be of significant therapeutic benefit for drugs having a narrow therapeutic index [5,6]. However, the skin comprising lipophilic stratum corneum (SC) and the aqueous living skin is a natural barrier that prevents drugs from entering the body. It is generally recognized that drugs pass through the skin by passive diffusion [7]. Therefore, appropriate solubility is helpful for the drug to be transported across the skin barrier. Unfortunately, Str is a drug with poor solubility both in water and lipophilic solvent.

**Figure 1 pharmaceuticals-15-00034-f001:**
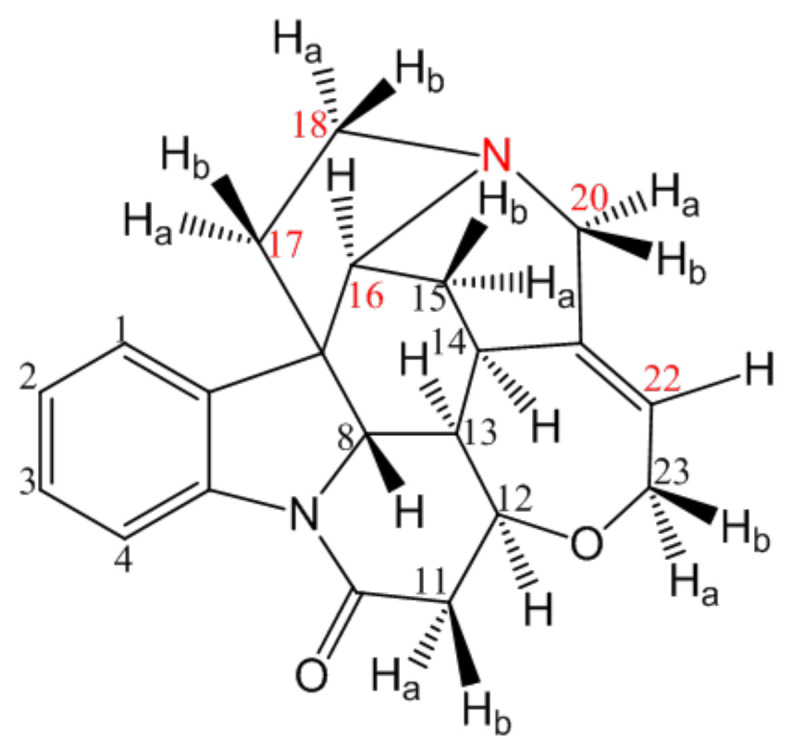
The chemical structure of Str.

The ion-pair strategy is popularly used to improve skin penetration by influencing the physicochemical properties of drugs, thereby avoiding changing the structure and pharmacological actions of the drugs [8,9]. It was reported that the skin permeability of ion-pair compounds was clearly influenced by physicochemical properties [10,11], stability [12], ionization in the viable epidermis [13] and the polar surface area [14]. Although the different reasons for the effects of ion-pair compounds on skin permeability were summarized in previous reports, it is still worth further exploring how ion-pair compounds affect the transdermal processes.

This study sought to investigate how Strychnine (Str) ion-pair compounds affect the in vitro transdermal process at the molecular level. Seven homologous fatty acids were selected to form ion-pair compounds with Str, and then the Spectroscopies of FTIR and NMR were employed to confirm these compounds. In vitro skin permeation experiments of Str and Str ion-pair compounds were also investigated. Then, the in vitro transdermal processes of Str and Str ion-pair compounds were explored in the Blends module in Material Studio 8.0 software (Accelrys, San Diego, CA, USA).

## 2. Results

### 2.1. ^1^H-NMR Spectra

^1^H-NMR spectroscopy could be used to study the phenomenon of proton transfer during the process of ion-pair formation [13,14]. The evidence for ion-pair formation between Str and different organic acids was provided by ^1^H-NMR spectroscopy, as shown in Figure 2. In addition, chemical shifts (δ, ppm) of Str and shift changes (Δδ, ppm) of Str compounds are separately listed in Table 1. The chemical shift values in H (16), H (17), H (18), H (20) and H (22) of the Str compounds increased. In summary, these downfield chemical shifts indicated that ion-pair compounds were formed between the Str and fatty acids.

### 2.2. FTIR Spectra

The Str ion-pair compounds were also confirmed by FTIR spectra [13,14], as shown in Figure 3. In general, the stretching vibration of the C=O group (υ C=O) of the saturated fatty acids ranged from 1714 cm^−1^ to 1689 cm^−1^, whereas the υ C=O of the Str was 1670 cm^−1^. When the Str reacted with the fatty acids, the υ C=O of C_10_, C_12_, C_14_, C_16_ and C_18_ were blue-shifted to 1722 cm^−1^~1716 cm^−1^. Compared with the υ C=O of Str, the υ C=O of C_4_ and C_6_ overlapped with the υ C=O of the Str, and a broad υ C=O absorption peak appeared. Moreover, the FTIR spectroscopy of the Str ion-pair compounds showed a new peak of 1567 cm^−1^~1556 cm^−1^, which was attributed to the absorption band of the COO^−^ group (ν COO^−^) between the –COOH of the fatty acids and N of the Str. Therefore, the proof of the formation of Str ion-pair compounds was also provided by the results of the FTIR spectroscopy.

### 2.3. Solubility and Apparent Partition Coefficient

The saturated solubility of the Strychnine and its ion-pair compounds in the phosphate buffer (pH 6.0) was more than 1600 μg/mL, as shown in Table 2. When the carbon atoms of the fatty acid were less than six, the Log K_O/W_ of the Str ion-pair compounds decreased significantly. However, the Log K_O/W_ of those compounds was similar to that of the Str when the carbon atoms of the fatty acid were more than 10.

### 2.4. The Skin Permeation of Str and Str Ion-Pair Compounds

As presented in Table 2 and Figure 4, the in vitro permeation fluxes of all Str ion-pair compounds were clearly promoted, and the 24 h cumulative amounts (Q_24h_) of the Str ion-pair compounds were 1.5–7.8-folds greater than that of the Str. The Str-C_10_ had the most significant enhancing effect, with Q_24_ of 857.54 ± 157.84 µg/cm^2^. These results showed that the Str ion-pair compounds had obvious effects on permeation fluxes.

### 2.5. Molecular Simulation

The hydrogen bond of the Str ion-pair compounds was also confirmed by molecular docking. As shown in Figure 5a–g, the hydrogen bond of the Str ion-pair compounds formed between the Str and all the selected fatty acids. In detail, the formation of deprotonated acid anions (RCOO^−^) and the protonated Str cation (^+^HNR) was due to the proton transfer between the -COOH of the fatty acids and the N_19_ of the Str. 

The Str and fatty acids were set to a base role (Ebb) and a screen role (Ess), respectively. Among these fatty acids, it was found that Capric acid (C_10_) had the best compatibility with Str, as shown in Figure 6. Water and the intercellular lipid were both set to a base screen role (Ebb), whereas the Str and Str ion-pair compounds were set to a screen role (Ess). As shown in Figure 7, the binding energy curve of the Str ion-pair compounds clearly changed, and their curves became more similar to the curves of the water and the intercellular lipid. When the binding energy curve of two substances is more similar, the compatibility of these substances is better [15]. Therefore, it was concluded that Str ion-pair compounds had better compatibility with the intercellular lipid and water than Str.

## 3. Discussion

The solubility of the Str was poor in water, normal saline and phosphate buffer pH7.4. In order to enhance the saturated solubility of the Str, phosphate buffer pH7.4: ethanol (8:2, *v*/*v*) was selected as the receptor solution in our previous experiment [16]. The saturated solubility of the Strychnine and its ion-pair compounds in a phosphate buffer (pH 6.0) was more than 1600 µg/mL because Str belongs to alkaloid with a pKa value of 8.29. The surface of normal skin is acidic, and the pH value of skin is 5.5 to 7.0 [17]. Therefore, a phosphate buffer (pH 6.0) was selected as the receptor solution. Isopropyl myristate is a kind of lipophilic solvent, which is frequently used as a vehicle solution in transdermal research [9,12]. Thus, isopropyl myristate was selected as the donor solution in this experiment.

The hydrogen bond of ion-pair compounds is a hydrogen bond with a proton transfer [18,19]. Organic acids and alkaloids can form hydrogen bonds in solvents of low dielectric constants [11,14]. The N_19_ of Str containing a lone electron pair can form a hydrogen bond with the proton of the chosen organic acid. Str has seven rings, which are not in the same plane because of the rigid structure. In the stereoscopic space, C_18_ and C_20_ were adjacent to N_19_, whereas C_16_, C_17_ and C_22_ neighbor N_19_. After forming the Str ion-pair compounds, the electron atmosphere density of those C atoms decreased for the electrostatic proximity effect of carbonyl oxygen. The hydrogen bond of the Str ion-pair compounds between the Str and the fatty acid was further confirmed by the molecular docking.

The hydrogen bond of ion-pair compounds is a weak interaction via coulomb attraction, which will break and form on an extremely short timescale [20]. The lifetime of ion-pair compounds (T_life_) was employed to assess the stability of the ion-pair compound. It was reported that the skin permeation of the ion-pair compound would increase with a longer T_life_ that was calculated by the spectrometer frequency and chemical shift changes [9,12]. After the Str ion-pair compounds were formed by Str and homologous fatty acids, the chemical shift changes of those compounds with more than ten carbon atoms were almost identical. According to those reports, it was concluded that the T_life_ of the Str ion-pair compounds should have the same value. Fortunately, it was found that the steady-state permeation flux of Str ion-pair compounds would increase if the compatibility of Str and the selected fatty acid in the process of molecular simulation were enhanced. Perhaps the T_life_ of Str ion-pair compounds would increase if the compatibility of Str and the selected fatty acid were enhanced.

The H-bonding number of penetrants had a pronounced effect on the diffusion coefficient of the SC [14,21]. The intercellular lipids have many donors and receptors of hydrogen bonds that could form a hydrogen bond with penetrants, which are important skin barriers for drugs [21]. The permeation flux of Zaltoprofen ion-pair compounds with alkylamines and cycloalkanolamines significantly increased due to the impairment of the SC with the carboxyl of Zaltoprofen, whereas the permeation flux of those compounds with alkanolamines significantly decreased because the additional H-bonding groups that were introduced increased the interaction with the SC [14]. In order to avoid introducing different functional groups to the ion-pair compounds, seven homologous fatty acids were selected to form ion-pair compounds with the Str.

The Log K_O/W_ is a key physicochemical property of drugs that has a close relationship with skin permeation, but the Log K_O/W_ could not always correctly reflect the results of permeation fluxes [22,23]. After forming ion-pair compounds, the permeation fluxes would increase for drugs with higher Log K_O/W_ [12,24]. As for drugs with appropriate Log K_O/W_, the permeation fluxes of ion-pair compounds would decrease [10,11]. When Str ion-pair compounds were formed by Str and fatty acids, the Log K_O/W_ of those compounds with fewer than six carbon atoms decreased significantly, whereas the Log K_O/W_ of those compounds with more than ten carbon atoms was close to the Log K_O/W_ of the Str. As the number of carbon atoms in the Str ion-pair compounds increased, their in vitro permeation fluxes increased step by step to the maximum value, and then decreased gradually. Therefore, there was a poor correlation between the Log K_O/W_ of the Str ion-pair compounds and their skin permeability fluxes.

It is popularly accepted that drug transport across the skin in vitro is controlled by a simple passive diffusion [25]. In fact, the driving force of passive diffusion is the chemical potential gradient, which is often simplified as the concentration gradient [25,26]. Fick’s first law of diffusion can usually be used to predict and analyze permeation data. According to the law, the steady-state permeation flux would increase with the skin-vehicle partition coefficient’s enhancement. Drugs passing through the skin in vitro must first be distributed from the vehicle to the skin. The higher the compatibility between the drugs and the skin, the more the drugs should be distributed to the skin from the vehicle. 

The binding energy of two components in a Blends simulation is an effective tool to distinguish their compatibility [15]. The property roles of the components can be distinguished in evaluating binding energies: one component acts as a base role (Ebb), and the other serves as a screen role (Ess). When the binding energy curve of two substances is more similar, the compatibility of these substances is better [15]. After the hydrogen bond of the Str ion-pair compounds was formed, their binding energy curve became more similar to the curves of the water and the intercellular lipid, which indicated that the affinity of the Str ion-pair compounds and skin was better than that of the Str and skin. It can be concluded that Str ion-pair compounds can be distributed from the vehicle to the skin more easily than Str. Therefore, the in vitro permeation fluxes of all the Str ion-pair compounds were significantly higher than those of the Str.

## 4. Materials and Methods

### 4.1. Materials

The Str was isolated and purified from Strychnos nux-vomica L. according to our previous report [27]. The content of Str was 95.7%, which was determined by HPLC. Stearic acid (C_18_), Palmitic acid (C_16_), Myristic acid (C_14_), Lauric acid (C_12_), Capric acid (C_10_), Caproic acid (C_6_), Butyric acid (C_4_), n-octanol and Isopropyl myristate were bought from Sinopharm Group Co. Ltd. (Shanghai, China). HPLC-grade methanol was purchased from Tianjin Kemiou Co. Ltd. (Tianjin, China). Other chemicals were of analytical grade.

### 4.2. Preparation of Str Ion-Pair Compounds

The Str ion-pair compound was prepared at room temperature according to previous reports [10,11]. As is standard, 2 g Str and fatty acids of equal molar ratios were dissolved in 100 mL chloroform and stirred with D-1 magnetic stirring (Gongyi, Henan, China) for 5 h. Then, chloroform was recovered by an RE-2000A rotary evaporator (Gongyi, Henan, China). Finally, the resulting ion-pair compounds were collected after being dried at 40 °C overnight by ZKXFB-1 vacuum drying (Shuli, Shanghai, China).

### 4.3. ^1^H-NMR Spectroscopy Studies

The samples were dissolved by deuterated chloroform (CDCl_3_) in NMR tubes for ^1^H-NMR analysis. The structures of the Str and Str ion-pair compounds were recorded by an AVANCE III HD 400 MHZ ^1^H-NMR spectrometer (Bruker, Karlsruhe, Germany). The chemical shifts were reported relative to tetramethylsilane (TMS.)

### 4.4. FTIR Spectroscopy Studies

The FTIR spectra of Str and Str ion-pair compounds were recorded on a Nicolet 6700 spectrometer (Thermo Fisher Scientific, Waltham, MA, USA) at room temperature. About 3 mg of samples were mixed with KBr, grinded and tableted. The spectra were obtained in a wave number region of 4000–400 cm^−1^ with a resolution of 4 cm^−1^.

### 4.5. Solubility Experiments

The solubility of the Str and its ion-pair compounds, both in a phosphate buffer pH 6.0 and isopropyl myristate, was carried out by adding excessive drugs to a 2.0 mL solvent in a sealed micro-vial. The samples were shaken at 32 °C in a water bath for 48 h, and then filtered by a membrane filter (0.45 μm, polytetrafluoroethylene). All samples were evaluated in triplicate. After being diluted by a methanol: water (25:75, *v*/*v*) solution with 1.0% formic acid and 0.3% triethylamine, in water when necessary, the samples were determined by HPLC.

### 4.6. Apparent Partition Coefficient Experiments

The shake-flask method was employed to determine the apparent partition coefficient (Log K_O/W_) of the Str and Str ion-pair compounds in water and *n*-octanol. *N*-octanol and distilled water were saturated with each other for 24 h before the experiment. All samples were sealed in a micro-vial and then shaken for 48 h at 32 °C. After centrifugation (10,000 rpm, 10 min), the sample concentration was determined by HPLC.

### 4.7. In Vitro Permeation Experiments

A porcine skin sample was prepared according to our previous report [16]. After the domestic pigs were killed in a local slaughterhouse (Xiangtan, China), the ears were immediately cut off. Then, the hair of the porcine ears was carefully removed with animal hair clippers, and the skin was harvested with a skin grafting knife (Medical Equipment Factory of Shanghai Medical Instruments Co., Ltd., Shanghai, China). The thickness of the skin was in a range of 0.6 ± 0.1 mm, and the integral skin samples were frozen at −20 °C until being used within two weeks.

Transdermal experiments were conducted at 37 ± 0.5 °C using a two-chamber Franz diffusion cell with a receiver volume of 6.5 mL and a valid diffusion region of 2.8 cm^2^. The prepared porcine skin was clipped between the diffusion cell and receptor cell, and the SC was facing the diffusion cell. Suspension of the Str and its ion-pair compounds in Isopropyl myristate was filled in the donor cell. The receptor cell was filled with a phosphate buffer (pH 6.0), which was put in a water bath and stirred with a magnetic bar at 300 rpm. At predetermined time intervals (i.e., 2, 4, 6, 8, 10, 12, 14, 24 h), 1 mL of receptor solution was collected for analysis, then 1 mL of fresh phosphate buffer (pH 6.0) was immediately added to the receptor cell. All experiments were repeated no fewer than three times, using porcine skins from different individuals.

### 4.8. HPLC Analysis

The samples were determined by an LC-20A HPLC system (SHIMADZU, Tokyo, Japan), which was equipped with an LC20A pump, an SPD-M20A diode array detector, a CTO-10AS column oven and an SIL-20A automated sampler. The separation was achieved on an InertSustain C_18_ column (4.6 mm × 250 mm, 5 µm, SHIMADZU, Tokyo, Japan) with the flow rate kept at 1.0 mL/min, the column temperature maintained at 40 °C and the mobile phase consisting of methanol: water solution (25:75, *v*/*v*) with 1.0% formic acid and 0.3% triethylamine in water. The run duration was fifteen minutes, and the retention time of the Str was 10.0 min. The injection volume for each run was 50 μL, and the detection wavelength was 254 nm. The linear equation of the Str was Y = 114,005X − 2340.1 (r^2^ = 0.999) at concentrations ranging from 0.0428 μg/mL to 10.7 μg/mL. The LOQ and LOD of the Str were 0.0428 μg/mL and 0.00856 μg/mL, respectively.

### 4.9. Molecular Simulation

The intercellular lipid of the SC is composed of ceramide, cholesterol and free fatty acid of equal molar ratios, and water is the most abundant molecule in the epidermis and dermis [28,29]. Therefore, the intercellular lipid was represented by ceramide, cholesterol and free fatty acid of a 1:1:1 ratio, and the hydrophilic viable skin was represented by water. At first, a smart algorithm (cascade of steepest descent, quasi-Newton methods and conjugate gradient) was employed to obtain the optimal geometry of the Str and the organic amine, respectively. Molecular docking was performed in the Blends module using Materials Studio Version 8.0 (Accelrys, San Diego, CA, USA), and COMPASS force field was conducted throughout the whole simulation process [13,14].

### 4.10. Data Analysis

The permeation profiles were presented by mapping the cumulative permeated amount per unit area (*Q*) versus time. The steady-state flux (*J*) was determined from the slope of the linear region of the permeation profile. All data were processed by Excel 2013 and reported as mean ± SD (*n* = 3).

In order to investigate the compatibility of the Str and its ion-pair compounds with skin, a molecular simulation was carried out in the Blends module of Materials Studio Version 8.0 (Accelrys, San Diego, CA, USA), and COMPASS force field was performed throughout the whole simulation process [15].

## 5. Conclusions

The in vitro permeation fluxes of all the Str ion-pair compounds were significantly higher than that of the Str. After the hydrogen bond of the Str ion-pair compounds was formed, their binding energy curve became more similar to the curves of the water and the intercellular lipid, which indicated that the Str ion-pair compounds had better compatibility with the intercellular lipid and viable skin than the Str. Therefore, it was concluded that Str ion-pair compounds can be distributed from vehicle to skin more easily than Str. These findings broadened our knowledge about how ion-pair compounds affect the skin permeation processes.

## Figures and Tables

**Figure 2 pharmaceuticals-15-00034-f002:**
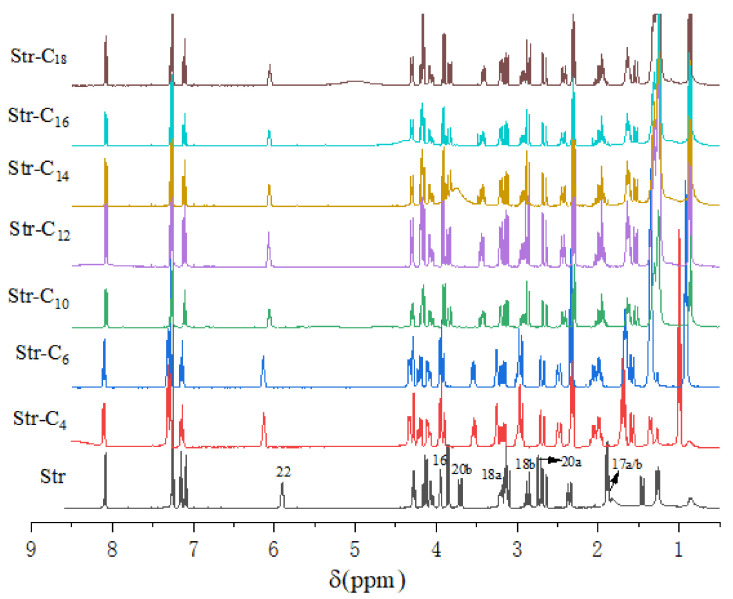
^1^H NMR spectra of Str and its ion-pairs in CDCl_3_.

**Figure 3 pharmaceuticals-15-00034-f003:**
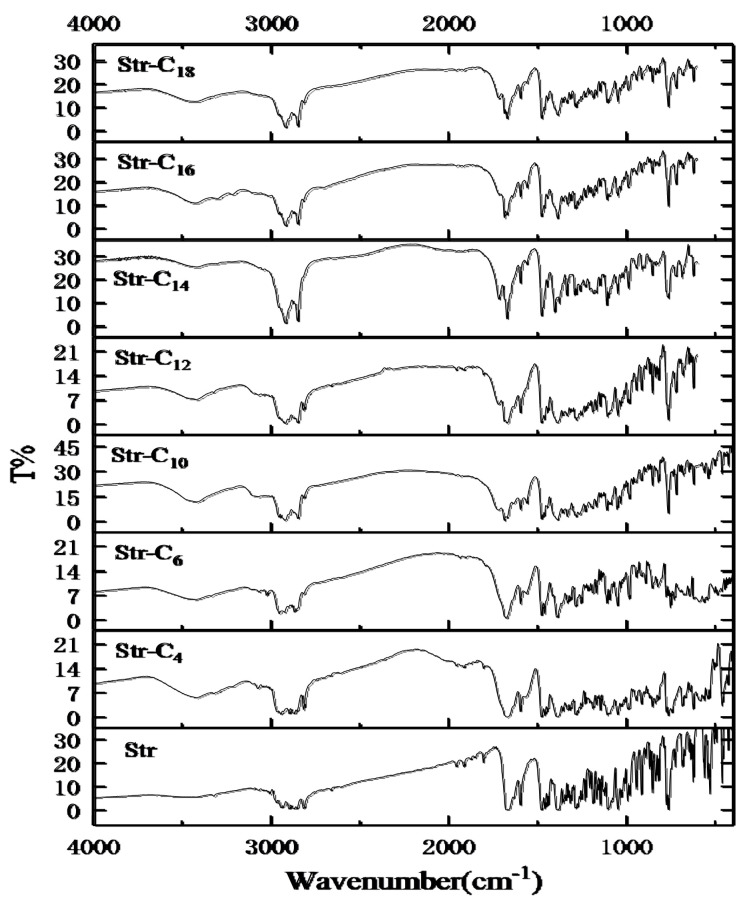
The FTIR spectra of Str and its ion-pair compounds.

**Figure 4 pharmaceuticals-15-00034-f004:**
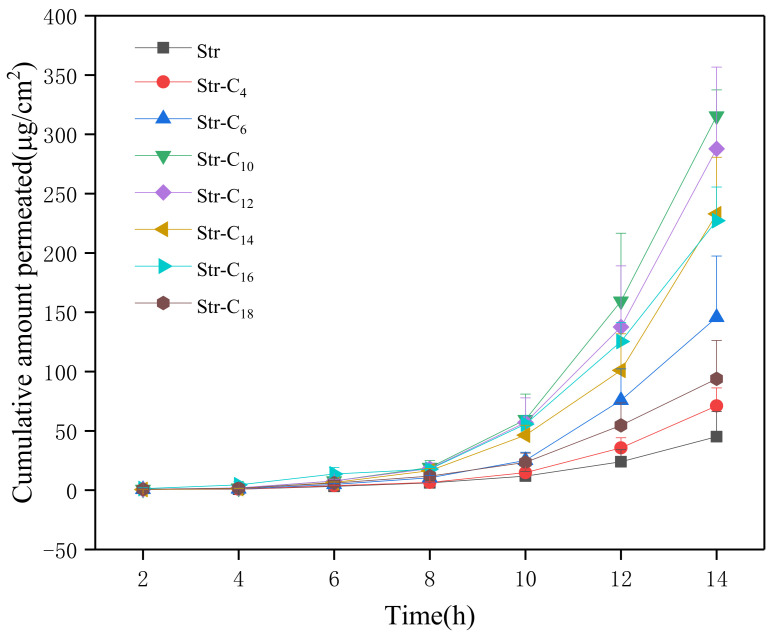
In vitro permeation profiles of Str and Str ion-pair complexes (*n* = 3).

**Figure 5 pharmaceuticals-15-00034-f005:**
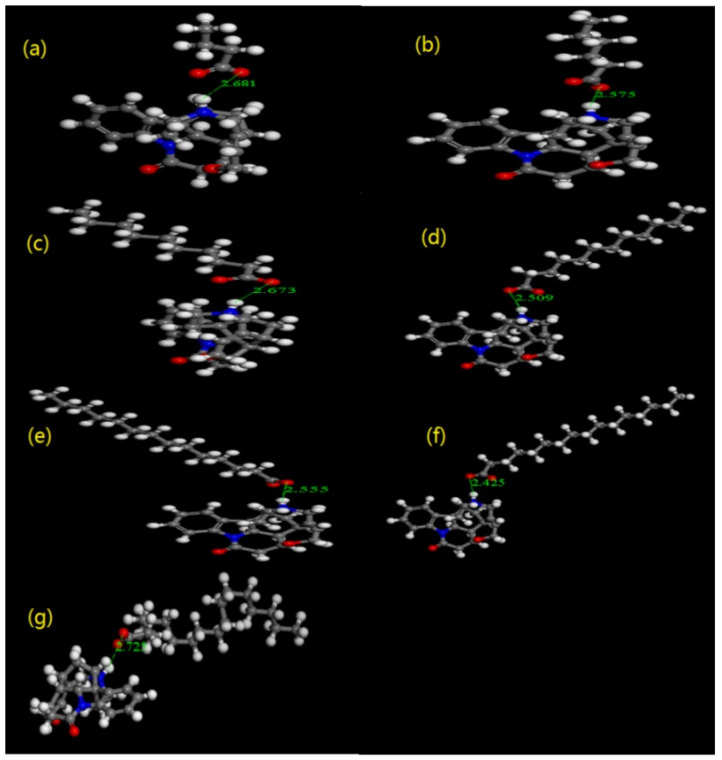
The molecular docking of Str (**a**) C_4_, (**b**) C_6,_ (**c**) C_10_, (**d**) C_12_, (**e**) C_14_, (**f**) C_16_, (**g**) C_18_.

**Figure 6 pharmaceuticals-15-00034-f006:**
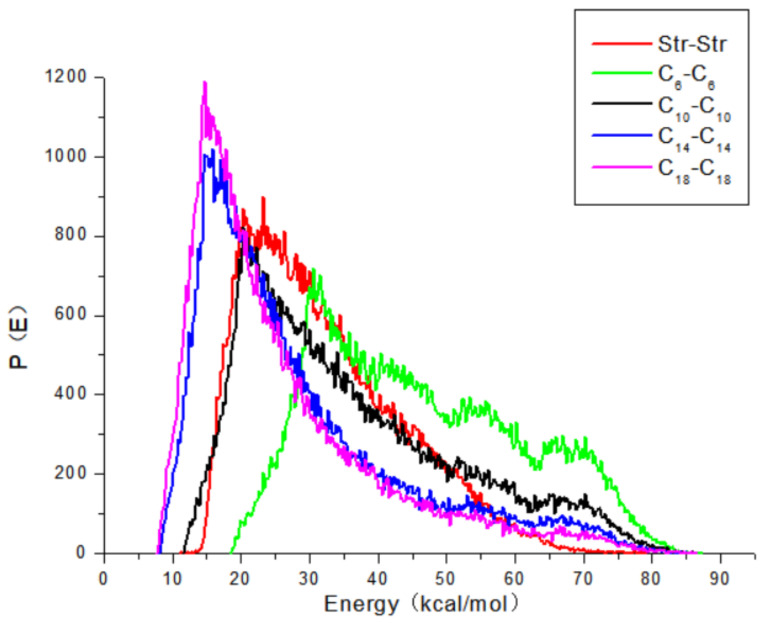
The binding energy distribution curves of Str and different fatty acids.

**Figure 7 pharmaceuticals-15-00034-f007:**
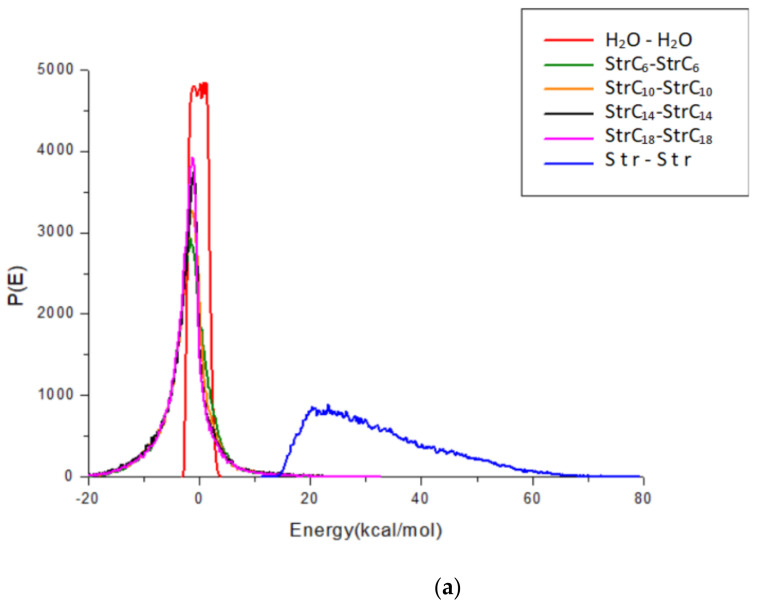
The binding energy distributions curves of (**a**) water and (**b**) ceramide, cholesterol and free fatty acid of equal molar ratios (COC) with Str and Str ion-pair compounds.

**Table 1 pharmaceuticals-15-00034-t001:** ^1^H NMR chemical shifts (δ, ppm) of Str and its ion-pairs in CDCl_3_.

	Str	Str-C_4_ Δδ	Str-C_6_ Δδ	Str-C_10_ Δδ	Str-C_12_ Δδ	Str-C_14_ Δδ	Str-C_16_ Δδ	Str-C_18_ Δδ
^22^H	5.92	0.2	0.21	0.14	0.15	0.14	0.14	0.13
^16^H	3.97	0.31	0.32	0.2	0.21	0.21	0.21	0.21
^20b^H	3.73	0.17	0.18	0.13	0.14	0.13	0.13	0.12
^18a^H	3.23	0.30	0.31	0.19	0.22	0.2	0.25	0.18
^18b^H	2.89	0.09	0.11	0.04	0.03	0.04	0.03	0.05
^20a^H	2.76	0.17	0.18	0.13	0.13	0.13	0.13	0.12
^17a/b^H	1.90/1.89	0.44/0.42	0.45/0.42	0.40/0.39	0.40/0.39	0.41/0.40	0.41/0.40	0.41/0.40

**Table 2 pharmaceuticals-15-00034-t002:** Skin permeation data of Str and its ion-pair complexes.

Samples	Q_24_ (µg/cm^2^)	*J* (µg/cm^2^/h)	T_lag_ (h)	S^a^ (μg/mL)	S^b^ (μg/mL)	Log K_O/W_
Str	101.24 ± 50.95	5.08 ± 2.39	6.02 ± 0.64	6579.5 ± 426.6	439.8 ± 53.4	1.18 ± 0.01
Str–C_4_	143.69 ± 34.18	8.18 ± 3.79	6.68 ± 0.34	8435.9 ± 30.4	422.5 ± 51.7	0.17 ± 0.02
Str–C_6_	475.17 ± 104.73	21.94 ± 10.84	6.76 ± 0.43	8996.4 ± 92.5	605.0 ± 131.2	0.61 ± 0.03
Str–C_10_	857.54 ± 157.84	42.79 ± 19.86	7.05 ± 0.17	4262.6 ± 173.5	556.6 ± 62.1	1.07 ± 0.01
Str–C_12_	789.88 ± 130.72	37.78 ± 13.72	6.88 ± 0.36	3090.1 ± 274.4	498.3 ± 34.1	1.16 ± 0.04
Str–C_14_	568.48 ± 112.79	27.51 ± 5.85	7.00 ± 0.12	1908.8 ± 123.5	420.1 ± 10.5	1.11 ± 0.02
Str–C_16_	521.09 ± 38.65	26.17 ± 10.2	5.85 ± 0.60	1701.9 ± 116.8	422.6 ± 8.3	1.07 ± 0.01
Str–C_18_	247.40 ± 82.73	10.93 ± 5.75	6.51 ± 0.58	1668.5 ± 261.8	433.2 ± 16.3	1.17 ± 0.02

S^a^ Solubility in phosphate buffer pH 6.0; S^b^ Solubility in Isopropyl myristate.

## Data Availability

Data is contained within the article.

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
