# Peer review of "Ion-Pair Compounds of Strychnine for Enhancing Skin Permeability: Influencing the Transdermal Processes In Vitro Based on Molecular Simulation"

_pharmaceuticals, 2021, doi:10.3390/ph15010034_

Round 1

Reviewer 1 Report

The manuscript presented by Lili He et al. and entitled “Ion-Pair Compounds of Strychnine for Enhancing Skin Permeability: Influencing on the Transdermal Processes in Vitro Based on Molecular Simulation” reports how ion-pair compounds, prepared between Strychnine and various fatty acids, could affect the in vitro transdermal process at molecular level.

Even though the aim of the study and the developed systems might be interesting, there are many scientific gaps that need to be addressed and some sections need to be completely rewritten with greater scientific rigor.

Furthermore, the entire manuscript is imperfect in terms of the writing style, the use of the English form and syntax is misleading, and some concepts are very difficult to read and understand. A deep proofreading is strongly encouraged.

Some observations are expressed in detail in the following comments:

Lines 29-36: The whole section is confused and hard to reed for the lack of prepositions and/or synonymous.  Form the very first sentence, “spinal cord” is over repeated and sounds redundant.

Line 32: in the context “which” is misleading.

Line 36: what the “fifteen” refer to? Minutes?

Line 37: reference of figure 1 through the text is missing, add it.

Line 41: Please, rewrite the sentence about the barrier activity of the skin.

Line 43: the sentence “Therefore it is helpful to overcome the barrier of skin that the drug has an appropriate solution” is misleading and the meaning is difficult to understand. Please improve it.

Line 50-51: How different changes of drug (already unclear concept) could be related to different effects of ion-pair compounds on skin permeability. Which are these overmentioned changes? It is not understandable also due to the absence of any reference.

Section 2.2: this section misses some important information about sample preparation. It is not clear the concentration of Str and fatty acids, the volume of chloroform employed, the model and manufacturer of magnetic stirrer, the size and materials of the used filters, how the temperature and the vacuum were kept overnight. Then a characterization in terms of yield (%) of the proposed method is totally absent. How could Authors prove that the method is effective and reproducible?

Lines 85-89: the statement “excessive drug” and the use of “solution” and “suspension” to point at the same subject are misleading and out of scientific context. Moreover, the material of filter membrane and the solvents where the analysis was carried out, are missing.

Line 92: what the number between brackets refers to? Moreover, n-octanol is missing in the materials paragraph, please add it.

Line 103: I would suggest adding a Standard error in the temperature because it is more probable that the experiment have been carried out in a temperature range. (i.e., 37 ± 0.5 °C).

Line 107: the medium with which the acceptor cell was filled is missing.

Line 110-111: the sentence is unclear.

Line 112-118: the description of HPLC analysis misses some information such as the name and type of modules (i.e., Column oven, pumps, detector, etc.). Moreover, the run duration is missing as well as the retention time of considered drug. Finally, calibration curve details, that I suppose Author used to determine the amount of Str, are missing as well as LOQ and LOD parameters.

Line 129-132: this part should be moved in the discussion paragraph; I suggest to Authors describe only the utilized method.

Lines 136-139: the whole part should be moved into the 2.7 paragraph. The Data analysis section should be reported the software used for data elaboration and for the statistical analyses. The latter is crucial to determine the significance of scientific data and must be added.

Line 168: the sentence “Thus phosphate buffer (pH6.0) was selected as the receptor solution” here is completely out of context since it refers to the permeation studies and here Authors should just present the results of Solubility and Apparent partition coefficient of Strychnine and its ion-pair compounds.

Line 169-170: “Isopropyl myristate is a kind of lipophilic solvent, which is frequently used as a vehicle solution in transdermal research [9,12].” Should be moved in discussion section.

Lines 170-172: the sentence is misleading, please improve it.

Line 192-195: this section would fit better in discussion section.

Line 194: The number of the figure is missing.

Line 217: figure 6 is not even mentioned along the text while figure 7, mentioned in the results section, is reported below. Thus, figure 7 must be moved there and fig 6 must be explained along the text.

Discussion: the entire section should be rewrite for many reasons. Firstly, the obtained results are not properly and really discussed, they must be presented with more scientifically details, such as the discussion about molecular simulation, that is totally unsatisfactory. Secondly, the syntax and the use of English are inadequate, thus the meaning of each statement is difficult to be understood and affect the possibility to well figure out the expressed concepts.

Conclusion: even though conclusions seem to be supported by results, these latter are not properly discussed and interpretated, thus it is hard to be completely aware of it.

Author Response

Even though the aim of the study and the developed systems might be interesting, there are many scientific gaps that need to be addressed and some sections need to be completely rewritten with greater scientific rigor. Furthermore, the entire manuscript is imperfect in terms of the writing style, the use of the English form and syntax is misleading, and some concepts are very difficult to read and understand. A deep proofreading is strongly encouraged. Some observations are expressed in detail in the following comments:

√Thank you very much for the valuable comments and suggestions that are helpful for us to improve this manuscript. Based on these comments and suggestions, we have made careful modifications to the original manuscript. We have checked the whole manuscript again and made several revisions. We hope that the revised manuscript is now suitable for publication.

Lines 29-36: The whole section is confused and hard to reed for the lack of prepositions and/or synonymous. Form the very first sentence, “spinal cord” is over repeated and sounds redundant.

√Thank very much for your suggestions. We had rewritten the first paragraph of the introduction.

Line 32: in the context “which” is misleading.

√We had corrected the sentence accordingly. “However Str is also known as a dangerous poison, which Str toxic dose is very close to its therapeutic dose”had been corrected as “However Str is also known as a dangerous poison that Str toxic dose is very close to its therapeutic dose”

Line 36: what the “fifteen” refer to? Minutes?

√Thank you for reminding us the missing word“minutes”. “Fifteen”refer to fifteen minutes.

Line 37: reference of figure 1 through the text is missing, add it.

√ We had added the reference of figure 1 in the first sentense of the first paragraph.

Line 41: Please, rewrite the sentence about the barrier activity of the skin.

√We had rewritten the sentences about the barrier activity of the skin accordingly. These sentences had be corrected as “However, the skin is a natural barrier that prevents drugs from entering the body. It is generally accepted that the skin is composed of lipophilic stratum corneum (SC) and the aqueous living skin, and that drugs pass through the skin by passive diffusion[7].”

Line 43: the sentence “Therefore it is helpful to overcome the barrier of skin that the drug has an appropriate solution” is misleading and the meaning is difficult to understand. Please improve it.

√We had improved this sentence,accordingly. This sentence had been corrected as “Therefore appropriate solubility is helpful for the drug to transport across the skin barrier. ”

Line 50-51: How different changes of drug (already unclear concept) could be related to different effects of ion-pair compounds on skin permeability. Which are these overmentioned changes? It is not understandable also due to the absence of any reference.

√Thank you for your suggestions.We had realized that the different changes of drug were unclear concept due to the absence of any reference.In our previously manuscript,the diverse changes referred to the above mentioned changes. We had added the reference “ Although the different effects of ion-pair compounds on skin permeability were ascribed to diverse changes of drug in the above reports”.

Section 2.2: this section misses some important information about sample preparation. It is not clear the concentration of Str and fatty acids, the volume of chloroform employed, the model and manufacturer of magnetic stirrer, the size and materials of the used filters, how the temperature and the vacuum were kept overnight. Then a characterization in terms of yield (%) of the proposed method is totally absent. How could Authors prove that the method is effective and reproducible?

√We had revised the progress of sample preparation,and supplemented some important information. “Str ion-pair compound was synthesized at room temperature. Typically, 2g Str and fatty acids of equal molar ratio were dissolved in 100ml chloroform and stirred with magnetic stirring(Gongyi,China) for 5 h. Then chloroform was recovered by rotary evaporator(RE-2000A,Gongyi, China). Finally,the resulting ion-pair compounds were collected after being dried at 40℃overnight by vacuum drying(ZKXFB-1,Shuli,China).”

    Str belongs to akaloid with the pKa value 8.29. Strychnine and fatty acids in chloroform belong to acid-base neutralization,and they should easily form salt. Thus the sentence “Str ion-pair compound was synthesized at room temperature”had been revised  to “Str ion-pair compound was prepared at room temperature”

Lines 85-89: the statement “excessive drug” and the use of “solution” and “suspension” to point at the same subject are misleading and out of scientific context. Moreover,the material of filter membrane and the solvents where the analysis was carried out, are missing.

√Thank you very much for careful review. We had revised our manuscript according to your comments.“Then the samples were shaken at 32 oC in water bath for 24 h. The suspension was filtered by a membrane filter (0.45 μm, polytetrafluoroethylene).After diluted by mobile phase when necessary”

Line 92: what the number between brackets refers to? Moreover, n-octanol is missing in the materials paragraph, please add it.

√Thank erry much for reminding us of this careless print error. We had deleted the mumber and brackets in this setence. The setence“Str ion-pair compounds in water and n-octanol (16)”had been corrected as “Str ion-pair compounds in water and n-octanol”. We had added n-octanol in the materials paragraph.

Line 103: I would suggest adding a Standard error in the temperature because it is more probable that the experiment have been carried out in a temperature range. (i.e., 37 ± 0.5 °C).

√We had added a Standard error in the temperature,accordingly.

Line 107: the medium with which the acceptor cell was filled is missing.

√ We had added the medium in this sentence,accordingly.The medium of the receptor cell was phosphate buffer(pH6.0).

Line 110-111: the sentence is unclear.

√The sentence of “All experiments were conducted with skin from different individuals in triplicate”had been revised as“All experiments were repeated no less than three times,using porcine skins from different individuals.”

Line 112-118: the description of HPLC analysis misses some information such as the name and type of modules (i.e., Column oven, pumps, detector, etc.). Moreover, the run duration is missing as well as the retention time of considered drug. Finally, calibration curve details, that I suppose Author used to determine the amount of Str, are missing as well as LOQ and LOD parameters.

√We had added the missing information of HPLC analysis accordingly.

The samples were determined by an LC-20A HPLC system (SHIMADZU,Japan), which equipped with an LC20A pump, a SPD-M20A diode array detector, a CTO-10AS column oven and a SIL-20A automated sampler.The separation was achieved on an InertSustain C18 column (4.6 mm × 250 mm, 5 µm, SHIMADZU, Japan) with the flow rate keeping at 1.0 mL/min, the column temperature maintaining at 40 °C and mobile phase consisting of methanol: water (25:75, v/v) with 1.0% formic acid and 0.3% triethylamine in water. The run duration was fifteen minutes, and the retention time of Str was 10.0 minutes. Injection volume for each run was 50 μL, and detection wavelength was 254 nm.The linear equation of Str was Y=114005X-2340.1 (r2 = 0.999) at concentrations ranging from 0.0428μg/ml to 10.7μg/ml. LOQ and LOD of Str were 0.0428μg/ml and 0.00856μg/ml,   respectively.

Line 129-132: this part should be moved in the discussion paragraph; I suggest to Authors describe only the utilized method.

√We had moved this part to the last paragraph of the discussion in the revised manuscript.

Lines 136-139: the whole part should be moved into the 2.7 paragraph. The Data analysis section should be reported the software used for data elaboration and for the statistical analyses. The latter is crucial to determine the significance of scientific data and must be added.

√We had moved this part to the Data analysis section in the revised manuscript.

“All dates were reported as mean ± SD”had been revised to“All data were processed by Excel 2013, and reported as mean ± SD. ”

Line 168: the sentence “Thus phosphate buffer (pH6.0) was selected as the receptor solution” here is completely out of context since it refers to the permeation studies and here Authors should just present the results of Solubility and Apparent partition coefficient of Strychnine and its ion-pair compounds.

√The sentence “Thus phosphate buffer (pH6.0) was selected as the receptor solution”had been moved  to the first paragraph of the discussion in the revised manuscript. Some contents unrelated to the results of solubility and oil-water partition coefficient were also moved to the discussion.

Line 169-170: “Isopropyl myristate is a kind of lipophilic solvent, which is frequently used as a vehicle solution in transdermal research [9,12].” Should be moved in discussion section.

√This sentence had been moved  to the second paragraph of the discussion in the revised manuscript.

Lines 170-172: the sentence is misleading, please improve it.

√The sentence“the Log KO/W of the Str ion-pair compounds with less than 6 carbon atoms significantly reduced. But the Log KO/W of those compounds with more than 10 carbon atoms were similar to the Log KO/W of Str.”had been revised to “When the carbon atoms of fatty acid were less than 6, the Log KO/W of the Str ion-pair compounds decreased significantly. However, the Log KO/W of those compounds were similar to that of Str when the carbon atoms of fatty acid were more than 10.” 

Line 194: The number of the figure is missing.

√Thank you for reminding us of  missing the key number of figure 6. The number of the figure was figure 6.

Line 192-195: this section would fit better in discussion section.

√This section was the explanation of figure 6.

Line 217: figure 6 is not even mentioned along the text while figure 7, mentioned in the results section, is reported below. Thus, figure 7 must be moved there and fig 6 must be explained along the text.

√ We regrettably missed the key number of figure 6, which had caused great trouble to the understanding of the a must rticle.The number of the figure was added in the revised manuscript. Figure 7 had been moved to the results section.

Discussion:the entire section should be rewrite for many reasons. Firstly, the obtained results are not properly and really discussed, they must be presented with more scientifically details, such as the discussion about molecular simulation, that is totally unsatisfactory. Secondly, the syntax and the use of English are inadequate, thus the meaning of each statement is difficult to be understood and affect the possibility to well figure out the expressed concepts.

√The discussion section has been proofread thoughly, and many revision had been marked on the revised manuscript. The discussion about molecular simulation had been rewritten at the last paragraph.This section had been revised as“Binding energy of two components in Blends simulation is an effective tool to distinguish their compatibility[19]. The role property of the components can be distinguished in evaluating binding energies: one component acts as a base role (Ebb) and the other serve as a screen role (Ess). When the binding energy curve of two substances is more similar, the compatibility of these substances is better [19].After the hydrogen bond of Str ion-pair compounds was formed, their binding energy curve became more similarly with those curves of water and the intercellular lipid, which indicated that the affinity of Str ion-pair compounds and skin was better than that of Str and skin. It could be concluded that Str ion-pair compounds could be distributed from vehicle to skin more easily than Str. Therefore,the in vitro permeation fluxes of all Str ion-pair compounds were significantly higher than those of Str.”

Conclusion: even though conclusions seem to be supported by results, these latter are not properly discussed and interpretated, thus it is hard to be completely aware of it.

√The results of the in vitro permeation fluxes and the compatibility of skin had been discussed at last paragraph in the revised manuscript.  The conclusion had been revised to “The in vitro permeation fluxes of all Str ion-pair compounds was significantly higher than that of Str. After the hydrogen bond of Str ion-pair compounds was formed, their binding energy curve became more similarly with those curves of water and the intercellular lipid, which indicated that Str ion-pair compounds had better compatibility with the intercellular lipid and viable skin than Str. Therefore it was concluded that Str ion-pair compounds could be distributed from vehicle to skin more easily than Str.”

Reviewer 2 Report

The work is interesting and well organized. The experimental part on ion pair characterization is well documented with 1H NMR, FTIR Spectroscopy studies and molecular simulation. The in vitro experiment with Franz cells must necessarily be repeated because there is a background error, the pH of the receptor compartment must be 7.4 because it mimic the human blood. For this reason most probably there is a bad correlation between the Log KO/W of Str ion-pair compounds and their skin permeability coefficient. So change the pH=7.4 in receptor compartment of Franz cells and after you can submitt again the paper.

Line 168: Thus phosphate buffer (pH6.0) was selected as the receptor solution

In your reference n. [16] they wrote: “The donor compartment was filled with donor solution and the receiver compartment with pH 7.4 phosphate buffer solution (PBS)”

Line 251 you wrote: “Therefore there was bad correlation between the Log KO/W of Str ion-pair compounds and their skin permeability coefficient”.

The error is in the pH of  Receptor compartment, it’s more acid than blood

Line 242: The Log KO/W is is a key physicochemica, there is two times is

Author Response

√Thank you very much for the valuable comments and suggestions that are helpful for us to improve this manuscript. Based on these comments and suggestions, we have made careful modifications to the original manuscript. We have checked the whole manuscript again and made several revisions. We hope that the revised manuscript is now suitable for publication. 

1.Line 168: Thus phosphate buffer (pH6.0) was selected as the receptor solution

In your reference n. [16] they wrote: “The donor compartment was filled with donor solution and the receiver compartment with pH 7.4 phosphate buffer solution (PBS)”

√The solubility of Str was poor in water, normal saline and phosphate buffer pH7.4[16]. In order to enhance the saturated solubility of Str, ethanol had been added to phosphate buffer pH7.4. In fact phosphate buffer pH7.4 : ethanol (8:2, v/v) was selected as the receptor solution in our previous experiment[16].       

The sentence“Thus phosphate buffer (pH6.0) was selected as the receptor solution” had been removed to the first paragraph of discussion in the revised manuscript.It had been discussed in the revised manuscript“The solubility of Str was poor in water, normal saline and phosphate buffer pH7.4. In order to enhance the saturated solubility of Str, phosphate buffer pH7.4 : ethanol (8:2, v/v) was selected as the receptor solution in our previous experiment[16]. The saturated solubility of Strychnine and its ion-pair compounds in phosphate buffer (pH 6.0) was more than 1600µg/ml because Str belongs to akaloid with the pKa value 8.29.Therefore, phosphate buffer (pH6.0) was selected as the receptor solution.”

2.Line 251 you wrote: “Therefore there was bad correlation between the Log KO/W of Str ion-pair compounds and their skin permeability coefficient” The error is in the pH of Receptor compartment, it’s more acid than blood

√Some concepts of this paragraph in the manuscripts were misleading in terms of the writing style, the use of the English form and syntax. We had revised them as “As for drugs with appropriate Log KO/W, the permeation fluxes of ion-pair compounds would decrease[10,11]. When Str ion-pair compounds were formed by Str and fatty acids, the Log KO/W of those compounds less than six carbon atoms decreased significantly, whereas the Log KO/W of those compounds more than ten carbon atoms was close to the Log KO/W of Str. As the carbon atoms number of Str ion-pair compounds increased, their in vitro permeation fluxes increased step by step to maximum value, and then decreased gradually. Therefore there was poor correlation between the Log KO/W of Str ion-pair compounds and their skin permeability fluxes.”

     The surface of normal skin is acidic, and pH value of skin is 5.5 to 7.0 [20]. (J. Zheng, P. Ding, L. Fang, New dosage form of transdermal administration, People's Medical Publishing House, 2006, 17.)  The solubility of Str was poor in phosphate buffer pH7.4,thus phosphate buffer pH7.4 : ethanol (8:2, v/v) was selected as the receptor solution of Str in our previous experiment[16]. The saturated solubility of Strychnine and its ion-pair compounds in phosphate buffer (pH 6.0) was more than 1600µg/ml. Therefore phosphate buffer (pH 6.0) were chosen as the receptor solution of Str in  our experiment.

Line 242: The Log KO/W is is a key physicochemica, there is two times is

√We had deleted one duplicate word “is”.

The in vitro experiment with Franz cells must necessarily be repeated because there is a background error, the pH of the receptor compartment must be 7.4 because it mimic the human blood. For this reason most probably there is a bad correlation between the Log KO/W of Str ion-pair compounds and their skin permeability coefficient.

√The in vitro experiment with Franz cells mimic the blood to take away drug immediately after the drugs pass through the skin,and the function of the receptor maintain the sink conditions of the receptor compartment.Therefore,the key factor of the receptor solution is the saturated solubility of drug. In the in vitro experiment of meloxicam, pH8.0 phosphate buffer were added to receptor compartment to maintain the sink conditions.( J. Drug Deli Transl. Res. 2018, 8, 64–72)  The solubility of Str was poor in pH7.4 phosphate, ethanol were added to pH7.4 phosphate buffer to enhance its saturated solubility[16]. Strychnine and its ion-pair compounds in phosphate buffer (pH 6.0) more than 1600µg/ml. Thus phosphate buffer (pH 6.0) were chosen as the receptor solution of Str in  our experiment.

As the carbon atoms number of Str ion-pair compounds increased, the Log KO/W of those compounds less than six carbon atoms decreased significantly, but the Log KO/W of those compounds more than ten carbon atoms was close to the Log KO/W of Str. However, their permeation fluxes increased step by step to maximum value, and then decreased gradually. Therefore there was poor correlation between the Log KO/W of Str ion-pair compounds and their skin permeability fluxes.

Reviewer 3 Report

The experimental design by the authors is scientifically interesting, the methods used to verify the skin permeability and the training of Str Ion-pair compounds are convincing.

The English form used has a good sound.

The bibliography includes only a couple of recent publications. A single note on the bibliography: the publications mentioned are obsolete, there is a single recent article.

Overall, the MS can be accepted.

Author Response

Thank  you very  much for your review comments.

Round 2

Reviewer 1 Report

I really appreciate the work that Authors made to improve the manuscript, especially in the discussion and conclusions sections, that are now more exhaustive and scientifically appropriate. However, some issues are still present, especially in the introduction and in methods sections, and an important characterization is still missing.

Thus, I suggest considering my answers and the further comments expressed below.

Even though the aim of the study and the developed systems might be interesting, there are many scientific gaps that need to be addressed and some sections need to be completely rewritten with greater scientific rigor. Furthermore, the entire manuscript is imperfect in terms of the writing style, the use of the English form and syntax is misleading, and some concepts are very difficult to read and understand. A deep proofreading is strongly encouraged. Some observations are expressed in detail in the following comments:

√Thank you very much for the valuable comments and suggestions that are helpful for us to improve this manuscript. Based on these comments and suggestions, we have made careful modifications to the original manuscript. We have checked the whole manuscript again and made several revisions. We hope that the revised manuscript is now suitable for publication.

Ok

Lines 29-36: The whole section is confused and hard to reed for the lack of prepositions and/or synonymous. Form the very first sentence, “spinal cord” is over repeated and sounds redundant.

√Thank very much for your suggestions. We had rewritten the first paragraph of the introduction.

Even though the paragraph was rewritten, some grammar errors are still present and the last part still remain confused.

Line 32: in the context “which” is misleading.

√We had corrected the sentence accordingly. “However Str is also known as a dangerous poison, which Str toxic dose is very close to its therapeutic dose”had been corrected as “However Str is also known as a dangerous poison that Str toxic dose is very close to its therapeutic dose”

Even “that” is still wrong. Here it is necessary the use of “since”.

Line 36: what the “fifteen” refer to? Minutes?

√Thank you for reminding us the missing word“minutes”. “Fifteen”refer to fifteen minutes.

Ok

Line 37: reference of figure 1 through the text is missing, add it.

√ We had added the reference of figure 1 in the first sentense of the first paragraph.

Ok

Line 41: Please, rewrite the sentence about the barrier activity of the skin.

√We had rewritten the sentences about the barrier activity of the skin accordingly. These sentences had be corrected as “However, the skin is a natural barrier that prevents drugs from entering the body. It is generally accepted that the skin is composed of lipophilic stratum corneum (SC) and the aqueous living skin, and that drugs pass through the skin by passive diffusion[7].”

In my opinion, it could still be improved.

Line 43: the sentence “Therefore it is helpful to overcome the barrier of skin that the drug has an appropriate solution” is misleading and the meaning is difficult to understand. Please improve it.

√We had improved this sentence,accordingly. This sentence had been corrected as “Therefore appropriate solubility is helpful for the drug to transport across the skin barrier. ”

Ok

Line 50-51: How different changes of drug (already unclear concept) could be related to different effects of ion-pair compounds on skin permeability. Which are these overmentioned changes? It is not understandable also due to the absence of any reference.

√Thank you for your suggestions.We had realized that the different changes of drug were unclear concept due to the absence of any reference.In our previously manuscript,the diverse changes referred to the above mentioned changes. We had added the reference “ Although the different effects of ion-pair compounds on skin permeability were ascribed to diverse changes of drug in the above reports”.

The concept of “diverse changes of drug” is still unclear and despite the changes made by the Authors, the whole sentence is still confused.

Section 2.2: this section misses some important information about sample preparation. It is not clear the concentration of Str and fatty acids, the volume of chloroform employed, the model and manufacturer of magnetic stirrer, the size and materials of the used filters, how the temperature and the vacuum were kept overnight. Then a characterization in terms of yield (%) of the proposed method is totally absent. How could Authors prove that the method is effective and reproducible?

√We had revised the progress of sample preparation,and supplemented some important information. “Str ion-pair compound was synthesized at room temperature. Typically, 2g Str and fatty acids of equal molar ratio were dissolved in 100ml chloroform and stirred with magnetic stirring(Gongyi,China) for 5 h. Then chloroform was recovered by rotary evaporator(RE-2000A,Gongyi, China). Finally, the resulting ion-pair compounds were collected after being dried at 40℃overnight by vacuum drying(ZKXFB-1,Shuli,China).”

    Str belongs to akaloid with the pKa value 8.29. Strychnine and fatty acids in chloroform belong to acid-base neutralization, and they should easily form salt. Thus the sentence “Str ion-pair compound was synthesized at room temperature” had been revised to “Str ion-pair compound was prepared at room temperature”

The missing information about preparation methods were added. However, the characterization in terms of yield (%) of the proposed method is totally absent and Authors explanation that “Str belongs to akaloid with the pKa value 8.29. Strychnine and fatty acids in chloroform belong to acid-base neutralization, and they should easily form salt” still do not prove that the method is effective and reproducible.

Lines 85-89: the statement “excessive drug” and the use of “solution” and “suspension” to point at the same subject are misleading and out of scientific context. Moreover, the material of filter membrane and the solvents where the analysis was carried out, are missing.

√Thank you very much for careful review. We had revised our manuscript according to your comments. “Then the samples were shaken at 32 oC in water bath for 24 h. The suspension was filtered by a membrane filter (0.45 μm, polytetrafluoroethylene).After diluted by mobile phase when necessary”

The solvent in which Authors determine the drug solubility is still missing. It is still unclear if Authors filtered all the suspension of just the supernatant. The sentence “After diluted by mobile phase when necessary” is useless if authors do not explain the mobile phase. Moreover, Authors write “the samples were repeated in triplicate” but, generally, are the ANALYSES to be repeated in triplicate, not the samples. And, in my opinion, these kinds of mistakes are inadmissible.

Line 92: what the number between brackets refers to? Moreover, n-octanol is missing in the materials paragraph, please add it.

√Thank erry much for reminding us of this careless print error. We had deleted the mumber and brackets in this setence. The setence“Str ion-pair compounds in water and n-octanol (16)”had been corrected as “Str ion-pair compounds in water and n-octanol”. We had added n-octanol in the materials paragraph.

Ok

Line 103: I would suggest adding a Standard error in the temperature because it is more probable that the experiment have been carried out in a temperature range. (i.e., 37 ± 0.5 °C).

√We had added a Standard error in the temperature,accordingly.

Ok

Line 107: the medium with which the acceptor cell was filled is missing.

√ We had added the medium in this sentence,accordingly.The medium of the receptor cell was phosphate buffer(pH6.0).

Ok

Line 110-111: the sentence is unclear.

√The sentence of “All experiments were conducted with skin from different individuals in triplicate”had been revised as“All experiments were repeated no less than three times,using porcine skins from different individuals.”

Ok

Line 112-118: the description of HPLC analysis misses some information such as the name and type of modules (i.e., Column oven, pumps, detector, etc.). Moreover, the run duration is missing as well as the retention time of considered drug. Finally, calibration curve details, that I suppose Author used to determine the amount of Str, are missing as well as LOQ and LOD parameters.

√We had added the missing information of HPLC analysis accordingly.

The samples were determined by an LC-20A HPLC system (SHIMADZU,Japan), which equipped with an LC20A pump, a SPD-M20A diode array detector, a CTO-10AS column oven and a SIL-20A automated sampler.The separation was achieved on an InertSustain C18 column (4.6 mm × 250 mm, 5 µm, SHIMADZU, Japan) with the flow rate keeping at 1.0 mL/min, the column temperature maintaining at 40 °C and mobile phase consisting of methanol: water (25:75, v/v) with 1.0% formic acid and 0.3% triethylamine in water. The run duration was fifteen minutes, and the retention time of Str was 10.0 minutes. Injection volume for each run was 50 μL, and detection wavelength was 254 nm. The linear equation of Str was Y=114005X-2340.1 (r2 = 0.999) at concentrations ranging from 0.0428μg/ml to 10.7μg/ml. LOQ and LOD of Str were 0.0428μg/ml and 0.00856μg/ml,  respectively.

Ok

Line 129-132: this part should be moved in the discussion paragraph; I suggest to Authors describe only the utilized method.

√We had moved this part to the last paragraph of the discussion in the revised manuscript.

Ok

Lines 136-139: the whole part should be moved into the 2.7 paragraph. The Data analysis section should be reported the software used for data elaboration and for the statistical analyses. The latter is crucial to determine the significance of scientific data and must be added.

√We had moved this part to the Data analysis section in the revised manuscript.

“All dates were reported as mean ± SD” had been revised to“All data were processed by Excel 2013, and reported as mean ± SD. ”

Ok

Line 168: the sentence “Thus phosphate buffer (pH6.0) was selected as the receptor solution” here is completely out of context since it refers to the permeation studies and here Authors should just present the results of Solubility and Apparent partition coefficient of Strychnine and its ion-pair compounds.

√The sentence “Thus phosphate buffer (pH6.0) was selected as the receptor solution”had been moved  to the first paragraph of the discussion in the revised manuscript. Some contents unrelated to the results of solubility and oil-water partition coefficient were also moved to the discussion.

Ok

Line 169-170: “Isopropyl myristate is a kind of lipophilic solvent, which is frequently used as a vehicle solution in transdermal research [9,12].” Should be moved in discussion section.

√This sentence had been moved to the second paragraph of the discussion in the revised manuscript.

Ok

Lines 170-172: the sentence is misleading, please improve it.

√The sentence“the Log KO/W of the Str ion-pair compounds with less than 6 carbon atoms significantly reduced. But the Log KO/W of those compounds with more than 10 carbon atoms were similar to the Log KO/W of Str.”had been revised to “When the carbon atoms of fatty acid were less than 6, the Log KO/W of the Str ion-pair compounds decreased significantly. However, the Log KO/W of those compounds were similar to that of Str when the carbon atoms of fatty acid were more than 10.” 

Ok

Line 194: The number of the figure is missing.

√Thank you for reminding us of  missing the key number of figure 6. The number of the figure was figure 6.

Ok

Line 192-195: this section would fit better in discussion section.

√This section was the explanation of figure 6.

Ok

Line 217: figure 6 is not even mentioned along the text while figure 7, mentioned in the results section, is reported below. Thus, figure 7 must be moved there and fig 6 must be explained along the text.

√ We regrettably missed the key number of figure 6, which had caused great trouble to the understanding of the a must rticle.The number of the figure was added in the revised manuscript. Figure 7 had been moved to the results section.

Ok

Discussion:the entire section should be rewrite for many reasons. Firstly, the obtained results are not properly and really discussed, they must be presented with more scientifically details, such as the discussion about molecular simulation, that is totally unsatisfactory. Secondly, the syntax and the use of English are inadequate, thus the meaning of each statement is difficult to be understood and affect the possibility to well figure out the expressed concepts.

√The discussion section has been proofread thoughly, and many revision had been marked on the revised manuscript. The discussion about molecular simulation had been rewritten at the last paragraph.This section had been revised as“Binding energy of two components in Blends simulation is an effective tool to distinguish their compatibility[19]. The role property of the components can be distinguished in evaluating binding energies: one component acts as a base role (Ebb) and the other serve as a screen role (Ess). When the binding energy curve of two substances is more similar, the compatibility of these substances is better [19].After the hydrogen bond of Str ion-pair compounds was formed, their binding energy curve became more similarly with those curves of water and the intercellular lipid, which indicated that the affinity of Str ion-pair compounds and skin was better than that of Str and skin. It could be concluded that Str ion-pair compounds could be distributed from vehicle to skin more easily than Str. Therefore,the in vitro permeation fluxes of all Str ion-pair compounds were significantly higher than those of Str.”

Ok.

Conclusion: even though conclusions seem to be supported by results, these latter are not properly discussed and interpretated, thus it is hard to be completely aware of it.

√The results of the in vitro permeation fluxes and the compatibility of skin had been discussed at last paragraph in the revised manuscript.  The conclusion had been revised to “The in vitro permeation fluxes of all Str ion-pair compounds was significantly higher than that of Str. After the hydrogen bond of Str ion-pair compounds was formed, their binding energy curve became more similarly with those curves of water and the intercellular lipid, which indicated that Str ion-pair compounds had better compatibility with the intercellular lipid and viable skin than Str. Therefore it was concluded that Str ion-pair compounds could be distributed from vehicle to skin more easily than Str.”

Ok .

Author Response

I really appreciate the work that Authors made to improve the manuscript,

especially in the discussion and conclusions sections, that are now more

exhaustive and scientifically appropriate. However, some issues are still present, especially in the introduction and in methods sections, and an important characterization is still missing.Thus, I suggest considering my answers and the further comments expressed below.

√Thank you very much for your further comments and suggestions about some issues in the introduction and in methods sections. We have revised the manuscript accordingly. Now we hope that the revised manuscript is now suitable for publication.

Lines 29-36: Even though the paragraph was rewritten, some grammar errors are still present and the last part still remain confused.

√Thank very much for your further comments. We had revised the last part of the first paragraph as “However, Str can excite the spinal cord reflex, cause tonic spasm and even lead to death caused by paralysis of the respiratory muscles [2,3].Since Str has a very narrow therapeutic index, it is also known as a dangerous poison in spite of its therapeutic values. Some patients  are prone to present poisoning symptoms during subcutaneous injection of clinical dose because the blood peak concentration of Str subcutaneous injection  would attain in fifteen minutes[2,4].”

Line 32: Even “that” is still wrong. Here it is necessary the use of “since”.

√We had corrected the sentence accordingly. “However Str is also known as a dangerous poison that Str toxic dose is very close to its therapeutic dose because Str can excite the spinal cord reflex, cause tonic spasm and lead to death for paralysis of the respiratory muscles [2,3]”had been corrected as “However, Str can excite the spinal cord reflex, cause tonic spasm and even lead to death caused by paralysis of the respiratory muscles [2,3].Since Str has a very narrow therapeutic index, it is also known as a dangerous poison in spite of its therapeutic values..”

Line 41: In my opinion, the sentences about the barrier activity of the skin could still be improved.

.

√We had improved the sentences about the barrier activity of the skin accordingly. These sentences had be corrected as “However, the skin comprising lipophilic stratum corneum (SC) and the aqueous living skin is a natural barrier that prevents drugs from entering the body. It is generally recognized that  drugs  pass through the skin by passive diffusion[7].”

Line 50-51: The concept of “diverse changes of drug” is still unclear and despite the changes  made by the Authors, the whole sentence is still confused.

√We  realize  that diverse changes of drug were confused concepts. We have tried to express it in another manner “ Although the different reasons for the  effects of ion-pair compounds on skin permeability had been summarized in the previous reports”.

Section 2.2:The missing information about preparation methods were added. However, the characterization in terms of yield (%) of the proposed method is totally absent. Authors still do not prove that the method is effective and Reproducible.

√In fact, we prepared  Str ion-pair compounds according to previous reports[10,11]. We regretted  missing these important literature in this section. Now  these literatures been added “Str ion-pair compound was prepared at room temperature according to previous reports[10-11].”

Lines 85-89: The solvent in which Authors determine the drug solubility is still missing. It is still unclear if Authors filtered all the suspension of just the supernatant. The sentence “After diluted by mobile phase when necessary” is useless if authors do not explain the mobile phase. Moreover, Authors write “the samples were repeated in triplicate” but, generally, are the ANALYSES to be repeated in triplicate, not the samples. And, in my opinion, these kinds of mistakes are inadmissible.

√Thank you very much for your careful review. Many details did not describe in the determination of drug solubility. The solvent of the drug solubility been added “The solubility of Str or its ion-pair compounds in phosphate buffer pH 6.0 and isopropyl myristate”. We have realized the ambiguous description about filtration, now it had been corrected to “The samples were shaken at 32 oC in water bath for 24 h, and then filtered by a membrane filter (0.45 μm, polytetrafluoroethylene).  We had added the diluted solvent “After being diluted by methanol: water (25:75, v/v) with 1.0% formic acid and 0.3% triethylamine in water ”. In fact, “All samples were repeated in triplicate” referred to the  drug solubility. Unfortunately, we did not realize when this sentence was misplaced. We had replaced this sentence“All samples were repeated in triplicate. After being diluted by methanol: water (25:75, v/v) with 1.0% formic acid and 0.3% triethylamine in water when necessary, ”

Reviewer 2 Report

The paper can be accepted in this form. 

Author Response

Thank you very much for your comments. 

Round 3

Reviewer 1 Report

I appreciate how the Authors followed the comments to improve the manuscript. However, there are still some lacks than need to be fixed and there are expressed in the comment below.

I really appreciate the work that Authors made to improve the manuscript, especially in the discussion and conclusions sections, that are now more exhaustive and scientifically appropriate. However, some issues are still present, especially in the introduction and in methods sections, and an important characterization is still missing. Thus, I suggest considering my answers and the further comments expressed below.

√Thank you very much for your further comments and suggestions about some issues in the introduction and in methods sections. We have revised the manuscript accordingly. Now we hope that the revised manuscript is now suitable for publication.

Ok

Lines 29-36: Even though the paragraph was rewritten, some grammar errors are still present, and the last part still remain confused.

√Thank very much for your further comments. We had revised the last part of the first paragraph as “However, Str can excite the spinal cord reflex, cause tonic spasm and even lead to death caused by paralysis of the respiratory muscles [2,3].Since Str has a very narrow therapeutic index, it is also known as a dangerous poison in spite of its therapeutic values. Some patients are prone to present poisoning symptoms during subcutaneous injection of clinical dose because the blood peak concentration of Str subcutaneous injection would attain in fifteen minutes[2,4].”

ok

Line 32: Even “that” is still wrong. Here it is necessary the use of “since”.

√We had corrected the sentence accordingly. “However Str is also known as a dangerous poison that Str toxic dose is very close to its therapeutic dose because Str can excite the spinal cord reflex, cause tonic spasm and lead to death for paralysis of the respiratory muscles [2,3]”had been corrected as “However, Str can excite the spinal cord reflex, cause tonic spasm and even lead to death caused by paralysis of the respiratory muscles [2,3].Since Str has a very narrow therapeutic index, it is also known as a dangerous poison in spite of its therapeutic values..”

Ok

Line 41: In my opinion, the sentences about the barrier activity of the skin could still be improved.

√We had improved the sentences about the barrier activity of the skin accordingly. These sentences had be corrected as “However, the skin comprising lipophilic stratum corneum (SC) and the aqueous living skin is a natural barrier that prevents drugs from entering the body. It is generally recognized that drugs pass through the skin by passive diffusion[7].”

ok

Line 50-51: The concept of “diverse changes of drug” is still unclear and despite the changes made by the Authors, the whole sentence is still confused.

√We realize that diverse changes of drug were confused concepts. We have tried to express it in another manner “ Although the different reasons for the effects of ion-pair compounds on skin permeability had been summarized in the previous reports”.

ok

Section 2.2:The missing information about preparation methods were added. However, the characterization in terms of yield (%) of the proposed method is totally absent. Authors still do not prove that the method is effective and Reproducible.

√In fact, we prepared Str ion-pair compounds according to previous reports[10,11]. We regretted missing these important literature in this section. Now these literatures been added “Str ion-pair compound was prepared at room temperature according to previous reports[10-11].”

The Yield (%) is still missing.

Lines 85-89: The solvent in which Authors determine the drug solubility is still missing. It is still unclear if Authors filtered all the suspension of just the supernatant. The sentence “After diluted by mobile phase when necessary” is useless if authors do not explain the mobile phase. Moreover, Authors write “the samples were repeated in triplicate” but, generally, are the ANALYSES to be repeated in triplicate, not the samples. And, in my opinion, these kinds of mistakes are inadmissible.

√Thank you very much for your careful review. Many details did not describe in the determination of drug solubility. The solvent of the drug solubility been added “The solubility of Str or its ion-pair compounds in phosphate buffer pH 6.0 and isopropyl myristate”. We have realized the ambiguous description about filtration, now it had been corrected to “The samples were shaken at 32 oC in water bath for 24 h, and then filtered by a membrane filter (0.45 μm, polytetrafluoroethylene). We had added the diluted solvent “After being diluted by methanol: water (25:75, v/v) with 1.0% formic acid and 0.3% triethylamine in water ”. In fact, “All samples were repeated in triplicate” referred to the drug solubility. Unfortunately, we did not realize when this sentence was misplaced. We had replaced this sentence “All samples were repeated in triplicate. After being diluted by methanol: water (25:75, v/v) with 1.0% formic acid and 0.3% triethylamine in water when necessary, ”

Thanks for the corrections. However, the sentence  “The solubility of Str or its ion-pair compounds in phosphate buffer pH 6.0 and isopropyl myristate was carried out …..” should be correct in ““The solubility of Str or its ion-pair compounds both in phosphate buffer pH 6.0 and isopropyl myristate was carried out …..”

Moreover, I would again suggest expressing as the analyses were repeated in triplicate and not the samples, because it is conceptually misleading, or alternatively "all samples were evaluated in triplicate”..

Author Response

I appreciate how the Authors followed the comments to improve the manuscript. However, there are still some lacks than need to be fixed and there are expressed in the comment below.

√Thank you very much for your careful review.We have learned a lot of writing skills from your review. We had revised the manuscript accordingly.

Section 2.2:The Yield (%) is still missing.

√Thank you for your comments. Str ion-pair compounds have a special type of hydrogen bond. The formation of hydrogen bond was due to the proton transfer from the -COOH to fatty acids and the N19 of Str. The hydrogen bond of Str ion-pair compounds was also confirmed by Fourier Transform Infrared (FTIR) Spectroscopy, Nuclear Magnetic Resonance (NMR) Spectroscopy and molecular simulation. The in vitro permeation fluxes of Str ion-pair compounds were 2.2 to 8.4 times that of Str.

Theoretically, Str and equimolar  fatty acids can completely form the hydrogen bond in lipophilic organic solvents. However,the hydrogen bond of ion-pair compound  breaks and forms on an extremely short timescale[9,12]. The hydrogen bond of ion-pair compound would ionize in water, and the content of ion pair compounds cannot usually be determined by HPLC. Therefore the yield (%) of ion-pair compounds was reported in previous literature[9-14]. If the mass of the product obtained is divided by the mass of strychnine and organic acid added, the yield of Str compounds was more than 90%.

Lines 85-89: Thanks for the corrections. However, the sentence  “The solubility of Str or its ion-pair compounds in phosphate buffer pH 6.0 and isopropyl myristate was carried out …..” should be correct in ““The solubility of Str or its ion-pair compounds both in phosphate buffer pH 6.0 and isopropyl myristate was carried out …..”

√Thank you very much for your wonderful suggestion.  We had revised accordingly. “The solubility of Str or its ion-pair compounds in phosphate buffer pH 6.0 and isopropyl myristate was carried out …..”had been corrected in “The solubility of Str or its ion-pair compounds both in phosphate buffer pH 6.0 and isopropyl myristate was carried out ….”

Moreover, I would again suggest expressing as the analyses were repeated in triplicate and not the samples, because it is conceptually misleading, or alternatively "all samples were evaluated in triplicate”.

√Thank you very much for your careful explanation. We realize that a word may have relatively fixed meaning.“All samples were repeated in triplicate”had been revised to "all samples were evaluated in triplicate”.

This manuscript is a resubmission of an earlier submission. The following is a list of the peer review reports and author responses from that submission.

Round 1

Reviewer 1 Report

I already reviewed this manuscript before and voiced serious concerns on several points. Now the authors resubmitted the article, but many of my concerns from the previous report have not been properly addressed in the second revision. In the present form, I still don't believe the article is suitable for publication in Pharmaceutics.

I will repeat the concerns from my last report which are still valid.  As I am a computational chemist, I will only focus on the computational part here.

(*) The authors state that they have performed a molecular simulation, but almost no computational details are given. In a research publication, it is a requirement that all computations and simulations must be reproducible by different software packages, and therefore, all details (simulation parameters, used algorithms, force field, ...) should be either named and cited from existing literature, or completely described in the manuscript or the supporting information.

(*) The authors state that they used a "smart algorithm" to find the minimum energy conformation of the ion pairs. What is this algorithm exactly? Was it published in literature before? If yes, please cite all relevant articles. If no, you need to describe the algorithm in all details. I am asking because it is a highly non-trivial problem to find minimum energy geometries of large molecules such as strychnine-fatty acid ion pairs. There are thousands of local energy minima, and it is very hard to ensure that the global energy minimum (or at least a relatively low minimum) has indeed been found. How is this question addressed in the optimization algorithm?

(*) The authors write that they used the COMPASS force field to describe the interactions. But no reference is given, and no further details are mentioned. A simple search did not reveal a COMPASS parameter set for strychnine. Please cite the articles in which the specific force field parameters for your molecules (strychnine and the fatty acids) are given, or please describe how you have obtained the force field parameters and report them (either in the manuscript or in the SI).

(*) The authors write that they have performed molecular simulation, but apart from a very short description of the protocol for the energy minimization of the initial structure, no details are mentioned. An energy minimization is per definition not a "simulation". What kind of simulation was performed? Was it force field molecular dynamics simulation? Was the solvent contained explicitly or via implicit solvent models? Were some bonds or angles constrained? Which time integrator algorithm was employed to solve the equations of motion? How was the time step chosen? Was there a thermostat active to keep the simulation temperature constant? Was the simulation performed under periodic boundary conditions to approximate the liquid phase? All these details are required in the manuscript (or SI) to allow for reproducibility, which is very important.

(*) What are the binding energy curves that are displayed in Figure 6 and 7? What is "P(E)" on the vertical axis? Despite being a computational chemist, I am not aware of this type of plot. A short explanation what is actually plotted here would be very helpful for the reader. Are these histograms over the encountered binding energies during the molecular dynamics simulations? If so, is it average pair-wise interaction between pairs of molecules? How does the "Blend" method for computing binding energies work? Some references to the literature would be good.

(*) The molecular structures in Figure 5 look quite distorted, and the Figure is of low quality. Please rework this Figure so that the structures can be properly identified.

Reviewer 2 Report

The resubmitted manuscript has improved. However, it seems that the authors didn't pay attention to many of the reviewers' comments/suggestions. Please kindly provide a point-by-point rebuttal letter.

- The references that the authors newly cited is not up-to-date and/or up to the international view, especially Reference no. 1 and 2. In addition, it should be "the People's of" in Reference no. 1. Also, what mean by [S], [M], [J] in Reference no. 1-4?

-"in vitro" should be italics

- For Table 2 results, the data variousation seems far too much (for example ± over 50 for some of the data set)